# Percutaneous Balloon Dilation in Two Dogs with Cor Triatriatum Dexter

**DOI:** 10.3390/vetsci9080419

**Published:** 2022-08-08

**Authors:** Valentina Patata, Tommaso Vezzosi, Giulia Calogero, Marta Croce, Helena Broch, Federica Marchesotti, Martina Bini, Oriol Domenech

**Affiliations:** 1Anicura Istituto Veterinario Novara, Strada Provinciale 9, Granozzo con Monticello, 28060 Novara, Italy; 2Department of Veterinary Sciences, University of Pisa, Via Livornese, San Piero a Grado, 56122 Pisa, Italy; 3Anicura Clinica Veterinaria CMV Varese, Viale Padre G.B. Aguggiari, 162, 21100 Varese, Italy

**Keywords:** ascites, coronary sinus, heart failure, congenital heart disease, cardiology, canine

## Abstract

**Simple Summary:**

This case report described in a detailed way the clinical presentation, management, and long-term follow up of a Rhodesian Ridgeback and an American Staffordshire Terrier affected by cor triatriatum dexter which successfully underwent percutaneous balloon dilation. Cor triatriatum dexter is a rare congenital heart disease in dogs characterized by the presence of a membrane that divides the right atrium into two chambers: a high-pressure caudal chamber and a low-pressure cranial chamber. Symptoms are present in around 60% of dogs with cor triatriatum dexter and medical treatment is generally not efficacious and surgical treatment is recommended. In this case series, both dogs were symptomatic and presented ascites without jugular venous distension or pleural effusion. Percutaneous balloon dilation was successfully performed, and both dogs had uneventful surgery recoveries. Moreover, the two dogs remain free of clinical signs and without cardiac medication three and three years and a half after the procedure.

**Abstract:**

Percutaneous balloon dilation was performed in a Rhodesian Ridgeback and in an American Staffordshire Terrier affected by cor triatriatum dexter (CTD). Both cases had ascites without jugular venous distension or pleural effusion. In both dogs the CTD presented a perforated membrane but with different morphology: in one case the coronary sinus entered the caudal chamber of the CTD together with the caudal vena cava. In the other case, the coronary sinus communicated with the cranial chamber of the CTD together with the cranial vena cava. Percutaneous balloon dilation of the CTD was successfully performed, and both dogs had uneventful surgery recoveries. At two years of follow-up, the dogs were free from clinical signs and cardiac medication.

## 1. Introduction

Cor triatriatum dexter (CTD) is a rare congenital cardiac defect of dogs accounting for 0.3% of all congenital heart diseases in this species [1]. It is characterized by the presence of an abnormal membrane that divides the right atrium into two chambers: a high-pressure caudal chamber (CaRA) and a low-pressure cranial chamber (CrRA) [2]. Cor triatriatum dexter derives from the persistence of the embryological Eustachio’s valve or embryologic right sinus venosus. This is a structure that has a physiological presence in embryonic life inside the right atrium, in order to direct oxygenated blood from the cranial vena cava, through the oval foramen, and into the left atrium. In some cases this structure fails to regress and causes abnormal right atrial chamber bipartition, impeding normal venous return from the caudal vena cava and coronary venous sinus [3]. This membrane can be imperforated or perforated. The size and number of the perforations underlie the clinical expression of this pathology. The imperforated forms or those characterized by small perforations will result, for patients at a very young age, in a typical pathological picture marked by signs of right-sided congestive heart failure [4,5].

In veterinary medicine, the currently described anatomical variants depend on the location of the coronary venous sinus outlet, which can take place at the level of the CaRA or the CrRA [6]. Echocardiography is the gold standard for CTD diagnosis since it can be used to visualize the pathological membrane, measure the orifice membrane, record the turbulent flow between the CaRA and CrRA, and evaluate concomitant congenital heart disease [7]. In severe forms, dilation of membrane perforation is indicated in order to reduce the pressure inside the CaRA.

There are several techniques for correcting CTD: balloon dilation, cutting and high-pressure balloon dilation, stent application, and membrane surgical resection in extracorporeal circulation or under total venous inflow occlusion [5,6,8,9,10,11,12,13,14]. However, only a few cases of CTD in dogs have been reported in the veterinary literature.

The aim of our case report was therefore to describe two cases of dogs affected by perforated CTD with detailed information on the history, clinical presentation, diagnostic imaging features, percutaneous balloon dilation, and post-interventional follow up.

## 2. Case Description

### 2.1. Case 1

A 2-month-old, 5 kg, female Rhodesian Ridgeback was referred to the Anicura Veterinary Clinic CMV (Varese, Italy) because of ascites associated with delayed growth compared with the other members of the litter. At physical examination, the dog was bright and alert, with a body condition score of 3/9, and presented a marked abdominal distension secondary to the severe ascites, without jugular distension or pulse. Moreover, the dog presented a mid-systolic murmur with an intensity of IV/VI and with a point of maximal intensity on the right side of the thorax.

Complete bloodwork was performed, which revealed mild hypoproteinemia (4.5 g/dL, normal values 5.2–8.2 g/dL) and mild hypoalbuminemia (2.1 g/dL, normal values 2.2–3.9 g/dL) compatible with the young age of the dog, and a moderate increase in alanine aminotranferase (304 U/L, normal values 10–125 g/dL) probably secondary to the hepatic venous congestion. Thoracic radiography was performed in two orthogonal views (right lateral and dorso-ventral) and showed mild right heart cardiomegaly with a dilated caudal vena cava and normal lung pattern. Electrocardiography revealed a sinus rhythm with increased amplitude of the P wave in lead II (0.5 mV, normal value < 0.4 mV) compatible with right atrial enlargement, and a right shift of the mean electrical axis of the QRS complex (150°, normal range 40–100°) compatible with an incomplete right bundle brunch block and/or right ventricular enlargement.

A complete echocardiographic examination was performed with the patient restrained on lateral recumbency, including 2-dimensional, mono-dimensional, and Doppler evaluation [15]. Right parasternal four-chamber-long axis view and right parasternal oblique views focused on right atrium and caudal vena cava showed the presence of a membrane dividing the right atrium into two chambers, a cranial and caudal one (Figure 1A,B). Membrane’s orifice was measured from the right parasternal oblique view focused on the right atrium and caudal vena cava using the blood tissue interface method (Figure 1B, Table 1). CaRA and CrRA were measured at their maximum diameter, excluding the coronary sinus, tracing a line parallel to the membrane from the right parasternal oblique view using the blood tissue interface method (Figure 1B, Table 1). Caudal vena cava diameter was taken from a subxiphoid longitudinal view with the dogs in right lateral recumbency measured at its widest diameter as previously described [16] (Table 1).

The low-pressure CrRA received the cranial vena cava and communicated with the tricuspid valve, while the high-pressure CaRA received the dilated caudal vena cava and the coronary sinus. The membrane was perforated by a single orifice with a diameter of about 3 mm (Table 1). The Doppler examination showed a continuous turbulent flow through the perforation (from the CaRA to CrRA) with a diastolic maximum peak velocity of 3 m/s during atrial systole (corresponding to a peak pressure gradient of 36 mmHg) (Figure 2A,B, Table 1).

The tricuspid valve leaflets were thickened, with incomplete coaptation and prolapse causing mild-to-moderate tricuspid regurgitation compatible with tricuspid valve dysplasia. The dimension and function of the left heart were within normal limits. Based on clinical and echocardiographic examination, a diagnosis of CTD associated with tricuspid valve dysplasia in right-sided congestive heart failure was made. Right-sided congestive heart failure was diagnosed based on the presence of ascites associated with caudal vena cava distension. Treatment with furosemide (1 mg/kg PO q12h), benazepril (0.25 mg/kg PO q12h), and spironolactone (2 mg/kg PO q24h) was started without success. Percutaneous balloon dilation of CTD was scheduled two weeks later once the minimal vessel diameter required for the procedure had been achieved in order to be able to use an introducer sheath of 8 Fr (3.5 mm of diameter of the right femoral vein). Written informed consents were obtained from the owners to perform the percutaneous balloon dilation. A preoperative complete coagulation profile was performed which included prothrombin time (PT) (8 s, normal values 6.9–10 s), activated partial thromboplastin time (aPTT) (15 s, normal values 13.2–20 s), and fibrinogen (215 mg/dL, normal values 125–400 mg/dL).

Percutaneous balloon dilation of the CTD was performed at the Anicura Istituto Veterinario Novara. Before the procedure, the severe abdominal effusion was partially drained in around 30 min to optimize ventilation during anesthesia. The dog was premedicated with methadone (0.2 mg/kg IV) and, after pre-oxygenation, general anesthesia was induced with fentanyl (3 mcg/kg IV), midazolam (2 mg/kg IV), and propofol (2 mg/kg IV). After orotracheal intubation, anesthesia was maintained with isoflurane (0.9% EtIso) in a mixed oxygen/air gas flow, and constant rate infusion of fentanyl (10 mcg/kg/h IV) was administered in order to provide analgesia and reduce the total requirement of isoflurane. The dog was positioned in right lateral recumbency. An 8 Fr × 11 cm sheath introducer (Boston Scientific, Marlborough, MA 01752, USA) was placed into the right femoral vein using a surgical cut down technique. Selective angiography was performed via a 5 Fr × 110 cm Pigtail catheter (Boston Scientific, Marlborough, MA 01752, USA) by injecting iohexol contrast into the caudal vena cava. This facilitated visualization of the CaRA and localization of the small perforation in the membrane separating the two chambers (Figure 3A).

A 0.035-inch straight-tip hydrophilic guidewire was advanced through the caudal vena cava and the CaRA. A 4 Fr × 65 cm Berenstein catheter (Infiniti Medical, Palo Alto, CA 94303, USA) was advanced along the guidewire. The CrRA was catheterized through the membrane perforation under fluoroscopic and transesophageal echocardiographic (TEE) guidance (transesophageal adult probe 5–7.5 MHz, Esaote TEE022, Multiplane probe). Subsequently the hydrophilic guide wire was exchanged with a Cook stiff guide wire 0.035 × 260 cm, the Berenstein catheter was removed, and all the balloon catheters were advanced through the stiff guide wire located in the right ventricle. Initially, a 12 mm × 4 cm long balloon catheter (Infiniti Medical, Palo Alto, CA 94303, USA) was introduced along the stiff guide wire and positioned to straddle the right atrial membrane which was gradually dilated. Three manual inflations were performed to distend the membrane perforation. The dilation procedure was then repeated using an 18 mm × 4 cm long balloon catheter (Infiniti Medical, Palo Alto, CA 94303, USA). The dimension of the balloons was based on the echocardiographic measurements of the length of the intra-atrial membrane and the diameter of the caudal vena cava, CaRA, CrRA, and of the tricuspid valve annulus. Post-dilation angiography and TEE confirmed an increased membrane perforation and increased flow from CaRA to CrRA in comparison with pre-dilation angiography and TEE (Figure 3B). The introducer was then removed, and the right femoral vein ligated.

The postoperative recovery was uneventful. Echocardiography showed a postoperative membrane perforation of approximately 10 mm with laminar flow through the orifice, and the diastolic peak flow velocity across the membrane significantly decreased to approximately 1.2 m/s (Figure 2C,D, Table 1). Mild-to-moderate tricuspid regurgitation due to tricuspid valve dysplasia remained unchanged. The dog was discharged without cardiologic medication. The ascites completely resolved within one week after the procedure. The dog was re-checked clinically and echocardiographically at 1, 3, and 6 months and once a year after the procedure for three years. The patient presented normal growth and remained free of clinical signs without cardiac medications during the entire follow-up period. The consecutive echocardiographic evaluations revealed an appropriate opening of the membrane’s orifice with a laminar flow through it, characterized by a slow diastolic velocity.

### 2.2. Case 2

A 2.8 year-old, 26 kg, castrated male American Staffordshire was referred to the cardiologic service of the Istituto Veterinario Novara (Novara, Italy) for exercise intolerance and ascites. At physical examination, the dog showed a poor body condition score (2/9) with evident sarcopenia and abdominal distension due to severe ascites without jugular vein distention or pulse.

Blood tests showed mild hypoalbuminemia (2.3 g/dL, normal range 2.5–4.4 g/dL). Electrocardiographic examination revealed a normal sinus rhythm. Thoracic radiography showed mild right heart cardiomegaly with a dilated caudal vena cava. A complete echocardiographic examination was performed as previously described including 2-dimensional, mono-dimensional and Doppler evaluations [15] which showed a pathological membrane at the level of the right atrium compatible with CTD. Concerning the echocardiographic measurements, the methods were the same used for case 1 (Table 2). The membrane was perforated with an orifice of 4 mm in diameter (Table 2). Doppler examination showed a turbulent flow through the membrane ostium, with a maximum peak velocity in diastole of 3.2 m/s (corresponding to a maximum pressure gradient of 40.9 mmHg) (Table 2). Unlike case 1, the coronary sinus and the cranial vena cava reached the CrRA, while a markedly enlarged caudal vena cava communicated with the CaRA (Figure 4).

Due to a marked abdominal effusion, a slow abdominocentesis (around 45 min) was performed and the percutaneous balloon dilation of the CTD was planned. A preoperative complete coagulation profile was performed, showing increased PT (120 s, normal values 6.9–10 s), increased aPTT (180 s, normal values 13.2–20 s), and marked fibrinogen reduction (24 mg/dL, normal values 125–400 mg/dL). A plasma transfusion was therefore performed preoperatively with normalization of the coagulation profile. For the percutaneous balloon dilation, the same anesthetic protocol as case 1 was used. The dog was positioned in right lateral recumbency, and the right femoral vein was isolated, allowing the placement of an 8 Fr × 11 cm sheath introducer (Boston Scientific, Marlborough, MA 01752, USA). Selective angiography was performed using a 6 Fr × 110 cm Pigtail catheter (Boston Scientific, Marlborough, MA 01752, USA), which revealed the location of the membrane’s perforation (Appendix A). A 6 Fr multipurpose catheter was then inserted over a 0.035-inch straight-tip hydrophilic guidewire and advanced through the membrane’s perforation in the CrRA. The hydrophilic guidewire was then replaced with a 0.0035″ 1.5 mm J-tipped 260 cm Extra-Stiff guide wire (Cook Medical, Bloomington, IN 47402-0489, USA) which was used to insert the balloon catheters. Three balloons of 10 mm × 4 cm, 14 mm × 4 cm, and 22 mm × 4 cm (Infiniti Medical, Palo Alto, CA 94303, USA) were inserted consecutively within the atrial membrane, thereby obtaining its progressive dilation (Video S2). As for case 1, the dimension of the balloons was based on the length of the Intra-atrial membrane, the diameter of the caudal vena cava, CaRA, and CrRA and the diameter of the tricuspid valve annulus.

A second angiography was performed to evaluate the membrane’s dilation and to demonstrate an increase in blood flow through the perforation (Video S3). The interventional procedure was carried out under fluoroscopy and transesophageal echocardiography guide.

The introducer was then removed, and the right femoral vein ligated. The dog recovered uneventfully after the procedure. Transthoracic echocardiography performed the day after revealed a 1 cm diameter orifice, with a laminar flow passing through the membrane with a significant reduction in its maximum peak velocity of 1.2 m/s (corresponding to a transmembrane pressure gradient of 5.7 mmHg) (Table 2). The ascites completely resolved within one week of the procedure. As for case 1, the dog was re-checked clinically and echocardiographically at 1, 3, and 6 months and once a year after the procedure for three and a half years. In the consecutive echocardiographic evaluations, the membrane’s orifice presented an appropriate opening with a laminar flow passing through it, which was characterized by a slow diastolic velocity. The dog was free of clinical signs without recurrence of ascites for the entire follow-up period.

## 3. Discussion

In dogs with CTD, the presence of symptoms depends principally to its anatomy and in particular on the presence or absence of communication between the two chambers (CaRA and CrRA) as well as with the size of the perforation of the membrane itself. With sufficiently large membrane perforations, a diagnosis of CTD may be accidental and there is no need for any therapeutic intervention [5]. In fact, it has been recently reported that around 40% of dogs with CTD do not need any treatment because the ostium is of a sufficient size to determine a low or no hemodynamic impact [5]. On the other hand, with one small membrane perforation, resistance to blood flow increases the pressure in the proximal chamber with direct consequences for venous return to the right heart. The resulting clinical syndrome is referred to by some authors as the “Budd–Chiari-like syndrome”. It is characterized by the development of ascites in the absence of pleural effusion and jugular distension and pulse [4]. In veterinary medicine, the most common morphological variant of CTD in dogs is characterized by the communication of both the caudal vena cava and coronary sinus outlet with the CaRA and the communication of the cranial vena cava with the CrRA, as in case 1 [5]. However, case 2 presented the coronary sinus entering the CrRA together with the cranial vena cava, while the caudal vena cava entered the CaRA alone.

Clinical signs of dogs affected by CTD are usually represented by the presence of ascites without pleural effusion and no jugular distension and pulsation. Pleural effusion is not a clinical feature of dogs with CTD because pleural drainage is guaranteed by venous circulation and lymphatic circulation. Both these systems, through the azygos vein, the internal thoracic vein, and the thoracic duct, reach the cranial vena cava and communicate with the low-pressure chamber, which is thus free from hemodynamic involvement [17]. Recently, a CTD with unusual morphology was described in a miniature schnauzer presenting jugular pulse and pleural effusion. The dog had both the cranial and caudal vena cava entering the CaRA, whereas the coronary sinus entered both CaRA and CrRA by two communications [18].

Case 1 had concurrent tricuspid valve dysplasia, which has been reported as the most common concurrent congenital abnormality in dogs with CTD, found in 53% of patients [3]. It is believed that the embryologic anomaly causing CTD is also involved in the development of tricuspid valve dysplasia. In fact, failure of apoptosis of the embryologic myocardial cells might induce both the incomplete separation of the tricuspid valve leaflets from the right ventricular wall and the persistence of the embryologic sinus venosus valve, conditions, which determined tricuspid valve dysplasia and CTD, respectively [19,20]. This mechanism could be a plausible explanation for the frequent correlation between the two congenital diseases [3]. However, CTD in dogs has been described together with other congenital diseases, such as pulmonary valve stenosis, mitral valve dysplasia, double-chamber right ventricle, patent ductus arteriosus, perimembranous interventricular septal defect, and left cranial vena cava persistence [5,13].

Case 2, differently to case 1, which had a normal coagulation profile, presented an increased PT and a PTT with a marked reduction in fibrinogen concentration. These alterations appear to be in line with hyperfybrinogenolysis, which is commonly reported in dogs with right-sided congestive heart failure [21]. Ascites in dogs and human has been suspected to be intrinsic fibrinolytic, and systemic fibrinolysis is suspected to be secondary to the absorption of the abdominal fluid rich in plasminogen activator, plasmin, or plasmin-like factors [21,22]. Although ascites could be considered as a potential cause of bleeding, this complication is rare in dogs with right congestive heart failure and spontaneous bleeding is not common [21].

Before the procedures in both dogs, abdominocentesis was performed. It is well demonstrated in humans as well as in dogs that severe ascites negatively affect respiratory function; therefore, it is advised to perform abdominocentesis in patients with severe abdominal effusion before anesthesia [23,24].

In the case of CTD presenting severe caudal vena cava congestion and ascites, interventional procedures are recommended because medical treatment is usually ineffective and does not relieve long-term symptoms due to the persistence of a mechanical obstruction to the flow [5]. Percutaneous balloon dilation of CTD is the most commonly used treatment for symptomatic dogs [5,6,9,25,26,27]. As in the present cases, this procedure is reported to be highly effective at the first attempt, and rarely presents complications [14,24,25]. Complications include membrane restenosis, failure to dilate the membrane ostium, and uncontrolled tearing of the atrial wall secondary to the fibromuscular nature of the membrane [8,14]. However, none of our cases presented these complications and no restenosis occurred during the 3-year follow-up period.

One alternative method is percutaneous balloon membranotomy using cutting balloon catheters [10]. Although this technique is reported as feasible and effective, it is limited by the availability of the small size of the cutting balloon and its high cost. Another technique is the application of an expandable stent through the membrane’s perforation [8,28]. A recent study reported the application of a stent for CTD palliation in six dogs, demonstrating that stent placement can be an effective alternative procedure to percutaneous balloon dilation. However, major intra-operatory and post-operatory complications were reported such as inappropriate device size, stent dislocation, and stent fracture [8]. Tricuspid regurgitation, commonly present in dogs with CTD, may potentially induce back and forth stent movement over the obstruction, thus weakening the stent matrix and facilitating stent fracture [8]. Moreover, stents are much more expensive than balloon catheters. However, stent positioning can be a valid procedure in dogs with CTD especially in the case of ineffective membrane dilation with balloon catheters. Lastly, a recent study reported the combination of membranostomy using torqueable biopsy forceps and balloon catheter dilation in a case of CTD restenosis [14]. Surgical approaches using extracorporeal circulation or total venous inflow occlusion are also described and generally used in cases of imperforated CTD [13].

## 4. Conclusions

This case report described the clinical features, therapeutic management, and long-term follow-up of two dogs affected by CTD. Percutaneous balloon dilation was successfully performed in both cases. The dogs were still asymptomatic over three years after the procedure, highlighting feasibility and long-term efficacy of percutaneous balloon dilation in dogs with CTD. This solution can be considered a first-line procedure in dogs with CTD given the low rate of complications, high efficacy, and low cost of the procedure.

## Figures and Tables

**Figure 1 vetsci-09-00419-f001:**
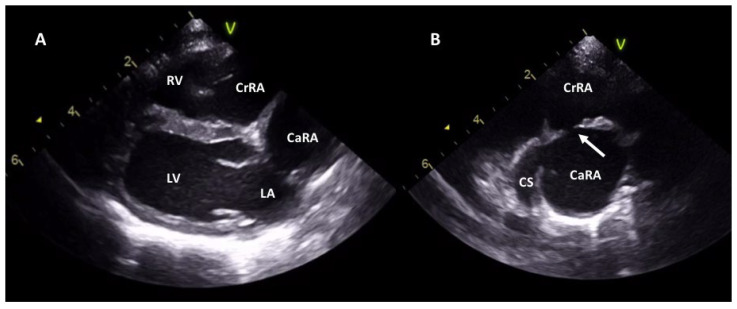
Two-dimensional echocardiographic images of case 1. Right parasternal long axis view (**A**), and right parasternal oblique view focused on the right atrium and caudal vena cava (**B**) showing the right atrium divided by a membrane into two chambers: the caudal chamber (CaRA) and cranial chamber. The two chambers communicate through a small perforation (white arrow). In figure B, we can notice the coronary sinus entering the CaRA. CaRA: caudal right atrium chamber; CrRA: cranial right atrium chamber; CS: coronary sinus; LA: left atrium; LV: left ventricle; RV: right ventricle.

**Figure 2 vetsci-09-00419-f002:**
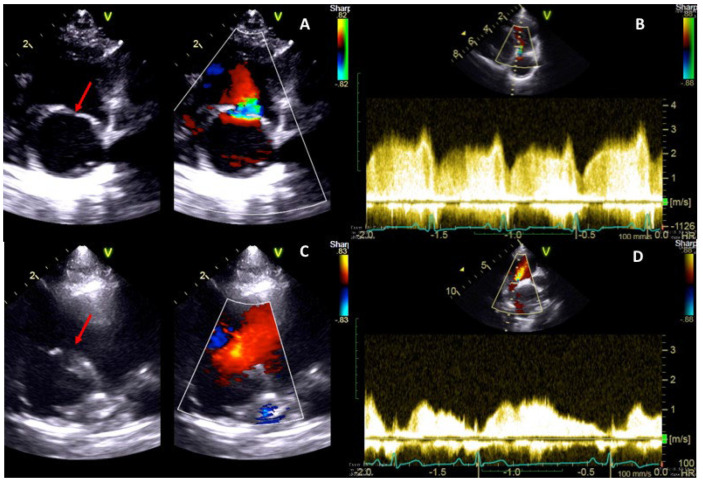
Two-dimensional, color, and spectral Doppler echocardiographic images of case 1. Right parasternal oblique view focused on the right atrium and caudal vena cava before (**A**) and after (**C**) the percutaneous balloon dilation of cor triatriatum dexter. After the procedure, the diameter of the perforation (red arrows) of the membrane increased markedly and a laminar flow could be seen through the membrane with color Doppler (**C**). Spectral Doppler images demonstrated a large reduction in the trans-membrane peak flow velocity (from 3.0 m/s to 1.2 m/s) (**B**,**D**). CaRA: caudal right atrium chamber; CrRA: cranial right atrium chamber.

**Figure 3 vetsci-09-00419-f003:**
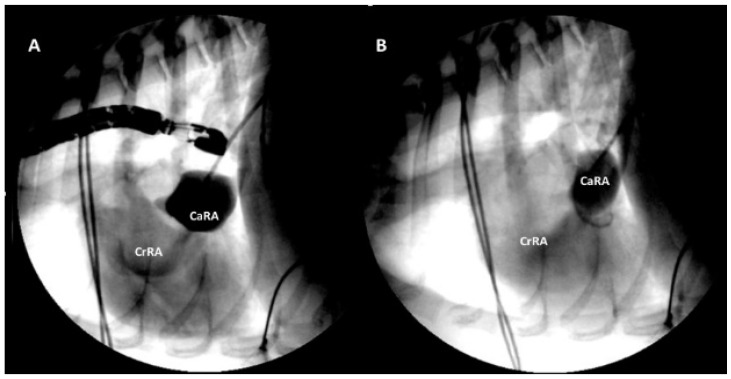
Lateral angiographic images of case 1 after contrast injection in the caudal vena cava pre-dilation (**A**) and post-dilation (**B**) showing increased size of the membrane perforation and increased flow from the caudal right atrial chamber to the cranial right atrial chamber.

**Figure 4 vetsci-09-00419-f004:**
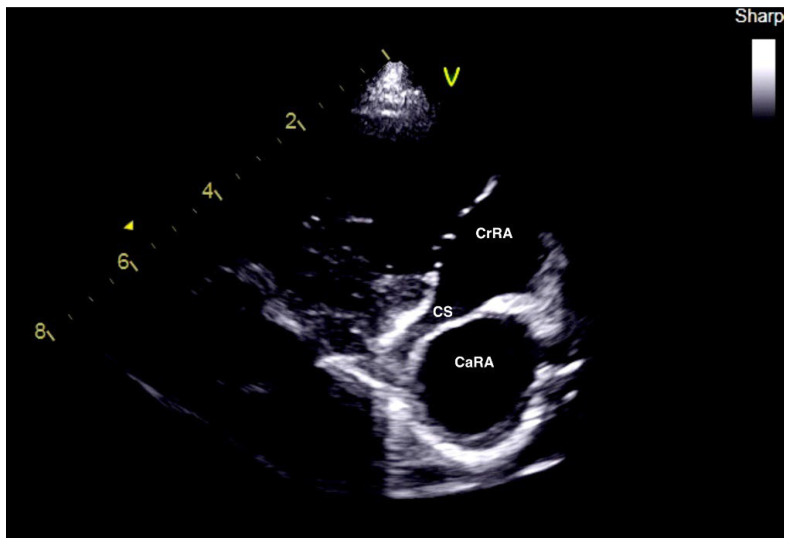
Two-dimensional echocardiographic images of case 2. Right parasternal oblique view focused on the right atrium and caudal vena cava (Figure 1B) showing the right atrium divided by a membrane in two chambers: the caudal chamber (CaRA) and cranial chamber (CrRA). Differently to case 1, the coronary sinus (CS) entered the CrRA. CaRA: caudal right atrium chamber; CrRA: cranial right atrium chamber; CS: coronary sinus.

**Table 1 vetsci-09-00419-t001:** Echocardiographic measurements of case 1 before and after the procedure.

	Before the Procedure	After the Procedure
Membrane’s orifice (mm)	3	10
Diameter CaVC (mm)	14.4	12.0
CaRA (mm)	23.1	15.5
CrRA (mm)	24.8	24.8
Tricuspid annulus (mm)	22.1	22.1
Transmembrane Vmax (m/s)	3	1.2

CaVC: caudal vena cava; CaRA: right atrium caudal chamber; CrRA: right atrium cranial chamber; Vmax: maximum peak velocity.

**Table 2 vetsci-09-00419-t002:** Echocardiographic measurements of case 2 before and after the procedure.

	Before the Procedure	After the Procedure
Membrane’s orifice (mm)	4	10
Diameter CaVC (mm)	19.5	17.0
CaRA (mm)	27.5	19.5
CrRA (mm)	26.8	26.8
Tricuspid annulus (mm)	27	27
Transmembrane Vmax (m/s)	3.2	1.2

CaVC: caudal vena cava; CaRA: right atrium caudal chamber; CrRA: right atrium cranial chamber; Vmax: maximum peak velocity.

## Data Availability

Not applicable.

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
