# Peer review of "Percutaneous Balloon Dilation in Two Dogs with Cor Triatriatum Dexter"

_vetsci, 2022, doi:10.3390/vetsci9080419_

Round 1
Reviewer 1 Report
This case report describes clinical features and therapeutic management with percutaneous balloon dilation in two dogs with cor triatriatum dexter.
The described approach, as well as the detailed description of the follow up, has been poorly described in veterinary medicine literature in this species till now. It’s very interesting to note that dogs described in this case report were still asymptomatic over three years after the procedure, highlighting long-term efficacy of percutaneous balloon dilation in dogs with this congenital defect with low rate of complications, high efficacy, and low cost of the procedure.
I have really appreciated the description of the procedures and the explanatory iconography attached.
Here are some minor comments:
Title:
“reCase” report is “Case report”? If yes, please, correct it.
Affialiations:
The submission guidelines require the initials of the name and surname of each author for each affiliation and the e-mail address. Please, verify it and add this information if lacking.
Introduction:
I believe that the introduction may be completed adding some prevalence data of this congenital heart disease in the dog population with their references.
Results:
- Line 69: according to the submission guidelines “All Figures, Schemes and Tables should be inserted into the main text close to their first citation and must be numbered following their number of appearance” – I therefore request to insert the figures and their captions in the text immediately after their citation and not at the end of the manuscript.
- Line 75: see previous comment.
- Line 98: see previous comment.
- Lines 115 and 120: see previous comment.
- Line 83: which parameters did you checked with the coagulation profile? For the case 2 (at lines 144-145) the specific results are reported. Can the authors insert the same information also for the case 1?
- Lines 93-93: “For the procedure, and a written informed consent was signed from the owner to perform the procedures.” I suggest removing this sentence because is a repetition of lines 81-82.
- Lines 142-143: the authors describe abdominocentesis for both cases. How long did they perform it?
- Figure 1 and Figure 2: please, verify the caption, particularly the explanation of high pressure caudal chamber (CaRA) and a low pressure cranial chamber (CrRA), because is reported “cranial right atrium chamber” for both the acronyms.
Discussion:
- Line 239: The authors discuss the coagulation profile results of the case 2. What about the results of the case 1? Can you add this information or discuss the reason of data lack?
- Lines 244-246: Does the literature report any data relating to the use of the abdominocentesis as a support to the treatment before the surgical procedure? If yes, please add this information in the discussion.

Author Response
The author would like to thank the reviewers for taking the time to review this manuscript. We found the comments appropriate and very useful. All points have been addressed. The changes to the manuscript are highlighted in yellow. The replies to the reviewer comment are reported in the Word file attached.

Reviewer 2 Report
Thanks for the invitation to review the manuscript of the case report entitled (Percutaneous balloon dilation in two dogs with cor triatriatum dexter).
In this report, the authors documented the surgical treatment of two dogs diagnosed with a congenital anomaly in the right atrium, cor triatriatum dexter.
The paper is interesting, however, significant editing of the manuscript is needed before any further processing.
significant rewriting is needed
i added many comments on the attached PDF

Author Response
The author would like to thank the reviewer for taking the time to review this manuscript. We found the comments appropriate and very useful. All points have been addressed. The changes to the manuscript are highlighted in yellow. The replies to the reviewer comment are reported in the PDF file attached.

Round 2
Reviewer 2 Report
Dear editor
thank you for sending me the paper again to revise. in the last time i carefully reviewed the paper, added some comments and suggestion to enhance the quality of the paper. some comments were corrected in the paper. However, many comments does not addressed. in addition, i would like to ask the authors to reply to each commnet separately in a document.
for example, in the last time, i suggested the following and i did not recieved the reply from authors. i know that the authors replied to some comments. but many comments no answer.
these are my previous comments
Thanks for the invitation to review the manuscript of the case report entitled (Percutaneous balloon dilation in two dogs with cor triatriatum dexter).
In this report, the authors documented surgical treatment of two dogs diagnosed with congenital anomaly in the right atrium, cor triatriatum dexter.
The paper is interesting, however, significant editing of the manuscript is needed before any further processing.
A table describing the echocardiographic findings before and after surgical treatment should be added with transorifice flow velocity. Even though the measurements were not significantly different from the reference range is considered a good reference for other studies. in addition, detailed description of echocardiographic approaches should be mentioned.
Line 50: change the heading to the Case description
Definitely describe the murmur. Describe the meaning of 4/6 is it the score of the murmur or the position of the murmur
Line 58-59: replace in numbers.
Describe the echocardiographic technique and measurements obtained before surgery in the two cases. Also, provide the follow-up measurements.
Line 201: which dogs and percentage that you refer to
The stage of heart failure should be fully described.
No description of medical treatment of case two. In addition, the treatment protocol after surgery is not mentioned.
Figures: You should provide echocardiographic images for each case separately with the same view and show the difference between the two cases. The arrangements of letters in figure 2 are not colorectal. No D and double C are found. I think a and b are before, C and D are after the operation
The discussion has many repeated sentences from the introduction. In addition, you compare the transcutaneous balloon and stent although you used balloon only. The discussion must be rephrased.
Line 22-26 and lines 193-197 are the same. Delete the repeated sentence in the discussion
other comments mentioned in the attached file
please provide step by step reply. for my previous comments and suggestions
thank you
Author Response
The author would like to thank the reviewers for taking the time to review this manuscript. We found the comments appropriate and very useful. All points have been addressed. The changes to the manuscript are highlighted in yellow. The replies to the reviewer comment are reported below.
